# Role and Limits of COVID-19 Vaccines in the Delicate Transition from Pandemic Mitigation to Endemic Control

**DOI:** 10.3390/vaccines10091555

**Published:** 2022-09-18

**Authors:** Marie Mura, Fabrice Simon, Vincent Pommier de Santi, Frédéric Tangy, Jean-Nicolas Tournier

**Affiliations:** 1Microbiology and Infectious Diseases Department, Institut de Recherche Biomédicale des Armées (IRBA), 91220 Brétigny sur Orge, France; 2Innovative Vaccine Laboratory, Institut Pasteur, 75015 Paris, France; 3Department of Infectious Diseases and Tropical Medicine, HIA Laveran, 13384 Marseille, France; 4French Armed Forces Center for Epidemiology and Public Health (CESPA), 13014 Marseille, France; 5École du Val-de-Grâce, 75005 Paris, France

**Keywords:** SARS-CoV-2, COVID-19, vaccine, variant, Omicron, Delta, pandemic, disease control

## Abstract

The recent surge of COVID-19 related to the Omicron variant emergence has thrown a harsh light upon epidemic control in the near future. This should lead the scientific and medical community to question the long-term vaccine strategy for SARS-CoV-2 control. We provide here a critical point of view regarding the virological evolution, epidemiological aspects, and immunological drivers for COVID-19 control, including a vaccination strategy. Overall, we need more innovations in vaccine development to reduce the COVID-19 burden long term. The most adequate answer might be better cooperation between universities, biotech and pharmaceutical companies

## 1. Introduction

The recent surge of COVID-19 related to the Omicron variant has thrown a harsh light upon pandemic control in the future. Recent studies report that the Omicron variant partially escapes immune protection induced by current vaccines and previous infections [1,2]. This should lead the scientific and medical community to question the long-term vaccine strategy for efficient control of this virus. The COVID-19 vaccine campaign massively started almost two years ago, in the winter of 2020–21, and was followed by a partial eclipse of the disease during the summer of 2021 (many other factors were at play, including non-pharmaceutical measures). However, most European countries were hit by the Delta variant, followed by the Omicron wave in the first semester of 2022. Most high-income countries have eased non-pharmaceutical interventions such as travel restrictions, physical distancing, face covering, hand washing, and closing of large events gathering during the first half of 2022. Nowadays, the COVID-19 vaccine effectiveness is considered the vaccine industry’s most exceptional and resounding success since its inception. In the World Health Organization (WHO) Europe region, it was estimated that between December 2020 and November 2021, the COVID-19 vaccination averted around half a million deaths [3]. The great winner of the world vaccine race was undoubtedly the mRNA vaccine technology [4,5]. Fast to be designed, scalable for a massive production, more or less stable at low negative temperature, easy to deliver, and efficient with only two shots, the mRNA vaccines were the solution that relegated their competitors as secondary competitors. As a result of the initial COVID-19 vaccine miracle, most people dreamt about a return to normal life [6].

One year later, the summer 2022 position is rather bitter with the emerging sublineages of Omicron variants BA.1 to BA.5, escaping previous immunity and potentially threatening most health care systems worn out by almost two years of struggles [7]. Hopefully, Omicron variants and its subtypes will induce a less severe disease than the previous Delta variant. Like Sisyphus rolling his stone at the foot of his mountain, ahead of us stands a potential overwhelming wave of new COVID-19 cases [8]. The world cannot carry on with a perspective of eternal containment, while boosting a perpetual waning immune protection of the entire population every four to six months. Furthermore, we still stay far from achieving a good and equitable global vaccine coverage. At this time, no real global strategy has been established for COVID-19 control, and we still stand far from disease control [9]. Very few countries have adopted a zero-COVID objective at the price of a very harsh confinement strategy [10], and no real public health objective has been assigned globally.

The scientific community has been at the forefront along the COVID-19 crisis for conceiving and developing novel counter-measures, and offering a panel of different vaccine strategies. Amidst this sharp scientific and industrial competition, it makes sense that pharmaceutical companies, governments, and medical authorities choose the most efficient vaccines available. With the Omicron subtype waves (and the future potential non-Omicron variant emergence) ahead of us, and the urgent need to get booster injections, we think that it is time now to question in depth the fundamental basis of current vaccine strategies.

## 2. Virological Concerns: Adapting to Viral Evolution Dynamic

The emergence of a pandemic respiratory agent observed with SARS-CoV-2 is recuring in the history of modern medicine. The last pandemic comparable in size in the 20th century was the 1918 influenza A (H1N1) virus that killed 50 to 100 million people worldwide across most continents with several deadly waves [11]. This was long before the modern virology era, and even far before the first identification of a virus by electron microscopy or the development of the first flu vaccine. Interestingly, the 1918 influenza A H1N1 pandemic virus survived hidden in animal or human reservoirs causing “pandemic bursts” in 1947, 1951, and 2009 [12].

For SARS-CoV-2, we have accumulated for the first time valuable phylogenetic data on the emergence and evolution of the virus and variants (more than 12.3 million complete sequences are referenced on GISAID, https://www.gisaid.org/ 15 September 2022). Such data provides a precise time-lapse of a phylo-geography evolution [13]. It is now clear that the SARS-CoV-2 is genetically drifting under evolutive constraints [14]. However, we do not precisely appreciate the SARS-CoV-2 evolutive capability, especially under constraint due to the pressure of global immunity. Most variants exhibit mutations in the spike protein, especially the receptor binding domain involved in its interaction with the cell entry receptor ACE-2. The receptor binding domain naturally represents a main antibody target for current vaccines [15]. The evolution of SARS-CoV-2 has been related to an improvement of fitness and infectiousness of the virus, but some variants also exhibit an immune escape phenotype, which seems particularly noticeable for the Omicron variant and its subtypes. The exact role of the immune pressure on the virus evolution is not really known [6]. A study has shown that the endemic human coronavirus 229E (hCo229E), responsible for the common cold, evolved progressively by mutating its spike protein. These accumulated alterations enabled the virus to escape neutralization within a decade of evolution [16].

SARS-CoV-2 has an animal reservoir origin in bats, and has been identified in a wide array of mammals [13,14,15,16,17], provoking reverse zoonosis, the most recent one being the wild white-tailed deer [18]. In the context of this large animal tropism, COVID-19 has become a health issue and the global aim of the vaccine campaign can never be the SARS-CoV-2 eradication [9]. The most ambitious aim could at best be taking control of the virus circulation in some geographical regions if we could get a vaccine blocking the transmission, before controlling its circulation more globally. From a long-term perspective, variant emergence would induce regular seasonal waves of infections.

## 3. Epidemiological Concerns: Improving the Global Access to the Current Vaccine

Although more than 11 billion vaccine doses have been administrated to date, epidemiological control of COVID-19 needs a better equity of the vaccine roll-out around the world, especially between wealthy countries and the poorest ones [19]. Recent data indicate that less than 15% of people in low-income countries have been (partially) vaccinated so far, leaving a staggering 2.7 billion people still to be vaccinated globally [20]. An infectious disease control strategy should be global, as we learned from the history of the smallpox eradication campaign [21], and the ongoing campaign for polio eradication led by the WHO [22]. It is interesting to notice that all SARS-CoV-2 emerging variants of concerns (VOC) so far arose from countries either before the initiation of their vaccine campaign (UK and Brazil for Alpha and Gamma VOCs, respectively), or with a low vaccine coverage (India for Delta VOC, and South Africa for Beta and Omicron VOCs) [13]. The globalization of the vaccine roll-out is thus not only an equity principle, but biosecurity insurance for every country.

We should also improve the strategies within countries with a large access to vaccines. Initially, the vaccine campaign was oriented to the most susceptible and fragile populations. However, we have artificially created a wedge between a fully immunized adult population and a completely naive child population. This was based on the observation that children were relatively spared from the disease and that the risk–benefit ratio was not favorable in those under 11 years old that develop very few severe forms of COVID-19. In the fall of 2021, the Delta variant caused a sharp rise in pediatric cases worldwide, while the incidence rate was fueling in ages 5 to 11, pushing the medical authorities to shift gears [23,24].

Overall, access to the vaccine should be globally improved, suggesting a need for a coordinated global effort and an improvement of the COVAX. Moreover, vaccine hesitancy represents an additional challenge to the lack of vaccine access in most countries [25].

## 4. Immunological Concerns: Improving the Future Vaccines

The vaccines developed so far display two major flaws: the rapid decline of the vaccine effectiveness within a few months [26], and a limited effect on host-to-host transmission [27].

First, more information about the mechanisms of vaccine protection against severe, symptomatic and pauci-symptomatic COVID-19 are eagerly awaited [28,29]. Indeed, waning antibody levels over time are suspected to be related to a renewed susceptibility to the infection. The humoral response, which is the easiest measurable endpoint, provides a global correlation of antibody level to early protection [30]; however, multiple factors are involved in the global immunity (neutralizing and non-neutralizing antibodies, CD8+ and CD4+ T lymphocytes) [28]. Beside the neutralizing capability of serum antibodies, the vaccine should trigger memory B and T cell responses that can mitigate the consequences of infection. Recent evidence suggests that mRNA vaccine BNT162b induces persistent follicular helper T cells that are crucial for the generation of long-lived plasma cells and memory B cells [31], and a long-term T cell immunity [32]. The exact role of the T cell response is not known, but it is suspected to protect against severe disease. More studies are needed to understand why vaccine immunity wanes over a few months, and how the memory T and B cell compartments could be stimulated more adequately. Moreover, a booster dose can improve vaccine effectiveness, although the duration of the booster effect is not yet known [29].

Noteworthy, most vaccines developed against respiratory tract infections induce a variable length of immune protection. Among them, it has been suggested that only viruses with systemic viremic spread induce a durable immunity (i.e., variola, measles, mumps, rubella viruses), but not viruses with a life cycle limited to the respiratory tract such as influenza viruses and coronaviruses [33]. Interestingly, the vaccinia virus induced a 10-year protection and was used in a post-exposure prophylaxis, while the measles vaccine induces a lifelong protection. As a comparison, the novel mRNA and adenovirus-based vaccines appear to be effective on infections for only a few months. It is urgent to know if this short length protection is correlated to the type of vaccine platform used or to the tropism of the virus. In this regard, other vaccine platforms already effective against SARS-CoV-2 in animal models, such as the measles vaccine [34,35] or the Ankara (MVA) [36,37] platform for vaccinia are of very high interest for further clinical trials.

Finally, it has also been suggested that COVID-19, like all other human coronaviruses, is progressively shifting from the status of an emerging agent circulating in a naïve population to an endemic virus propagating more laboriously in a population previously infected or vaccinated [38]. Because virtually everyone experiences multiple exposures through one or more vaccine doses or virus contacts, the disease may progressively fade into an array of milder symptoms [39]. This is maybe what is observed with the Omicron variant nowadays, although it is not known if this is due to an intrinsic property of the variant [40].

Transmission control also points out the need for a better oriented immune response to the mucosal immune system [27]. Little is known on how to improve lung immunity, although lung infections represent one of the deadliest infectious diseases worldwide. Live attenuated vaccines, although administered intramuscularly, elicit long-term mucosal protection against respiratory-acquired infections (measles, mumps, smallpox) because they spread systemically during their replication. All SARS-CoV-2 vaccines currently approved are injected intramuscularly, which may not be the optimum route of administration for mucosal education. In the absence of live-attenuated vaccines against SARS-CoV-2, the local administration of existing vaccines should be tested to improve the local mucosal protection and to sterilize the transmission. Some vaccines administered intranasally have been tested with success in animal models [36,41,42]. It is undoubtedly a promising way of administrating vaccines and for the generation of tissue-resident memory T cells [43].

## 5. From Vaccines to a Vaccination Strategy

Eradication of SARS-CoV-2 is impossible and herd immunity seems to be a pipe dream for the epidemic control. As for any emergence, the choice is still difficult between protecting the national health care system, reducing the incidence of severe forms, and/or preserving national economic interests. The survival of health systems remains the main driver for vaccine strategy today. It implies continuing efforts to vaccinate the whole population, especially those most vulnerable. The strategic choices are limited to convincing, coercing, and making vaccination compulsory. A strategy with repeated vaccine boosters for an entire population does not seem sustainable, even in high-income countries. Considering the increasing burden of breakthrough infections, the mRNA vaccine is not the Holy Grail, but rather an appropriate technology for an urgent response to a pandemic alert. This in turn affords time for slower developing, yet more efficacious, vaccines to access the market. Each country defines its own health strategy, like that for influenza vaccination in Europe with various targets (protecting the vulnerable, protecting healthy children, adolescents, and adults, health care workers) [44]. The solution would arise from comparing these different strategies (target populations, vaccines) to identify those most effective. For example, heterologous vaccine strategies (ChAdOx1-S-nCoV-19 and BNT162b2 combination) have been shown to be more efficient than homologous BNT162b2 and BNT162b2 combination in a real-world observational study of healthcare workers [45].

For the first time during a pandemic, some high-income countries have had access to national health databases, including COVID 19 diagnoses and patient outcomes. This allows a real-time assessment of the disease burden, using summary measures of population health like disability-adjusted life years (DALYs) due to both acute and long-lasting morbimortality. Better than incidence and infection fatality rates, DALYs could provide comprehensive and comparative public health information to assess and guide decision-making on vaccine strategy [46,47].

## 6. Conclusions

The resolution of the COVID-19 pandemic sounds more complex than initially thought. The fight against a novel virus that infects the respiratory tract and is transmitted to a naïve population is a daunting task that needs global reactivity and innovation. With the Omicron variant we may be entering a transition from pandemic to an endemic phase, with a lower death toll. It will take many years to deal with this newcomer, as evidenced by our experience with measles control and elimination despite more than 50 years of massive vaccination with one of the most efficient live attenuated vaccines. Success towards endemic control of SARS-CoV-2 or any other emerging agent will result from a better synergy between the scientific community, big pharma, decision makers, and health authorities, all driven by a strong public health spirit.

## Data Availability

Not applicable.

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
