# Peer review of "Role and Limits of COVID-19 Vaccines in the Delicate Transition from Pandemic Mitigation to Endemic Control"

_vaccines, 2022, doi:10.3390/vaccines10091555_

Round 1

Reviewer 1 Report

Manuscript Vaccines -1887241

Title: COVID -19 vaccines: a delicate transition from pandemic mitigation to endemic control

Authors:  Marie Mura 1,4, Fabrice Simon 2, Vincent Pommier de Santi 3, Frédéric Tangy 4, and Jean-Nicolas Tournier 1,4,5,* 

The present article is interesting and well written and provides an analysis of the virological, epidemiological and the efforts for establishing a disease control of COVOD-19 infection. The authors describe the actions that should be done to control the disease and the need to address a vaccination strategy to control the disease and the  need of  more innovations to reduce the 19 COVID-19 burden on the long term, and the most adequate answer would emerge from a better 20 cooperation between universities, biotech and pharmaceutical companies.

 The recent surge of COVID-19 related to the omicron variant emergence has thrown a harsh light 14 on the expectation of the epidemic control in the near future. This should lead the scientific and 15 medical community to question on the long-term vaccine strategy to deploy for an efficient global 16 SARS-CoV-2 control. We provide here an analysis of the virological, epidemiological, and immuno-17 logical drivers of the current global COVID-19 issues for establishing a disease control, and the 18 means to address a vaccination strategy control. Overall, we need more innovations to reduce the 19 COVID-19 burden on the long term, and the most adequate answer would emerge from a better 20 cooperation between universities, biotech and pharmaceutical companies

Author Response

We appreciate the reviewer comments. As suggested by the reviewers, we have substantially modified our manuscript and provide below an answer to all comments.

Reviewer #1 (Comments for the Author):

The present article is interesting and well written and provides an analysis of the virological, epidemiological and the efforts for establishing a disease control of COVOD-19 infection. The authors describe the actions that should be done to control the disease and the need to address a vaccination strategy to control the disease and the need of more innovations to reduce the 19 COVID-19 burden on the long term, and the most adequate answer would emerge from a better 20 cooperation between universities, biotech and pharmaceutical companies.

We appreciate the reviewer comments and thank the reviewer for his consideration.

Reviewer 2 Report

The results listed in the paper in the form of formulas, figures, and analysis seems true and correct. The paper is well written and it is written in a truly sporty manner. English is generally good, I think it needs to be polished further and some typos need to be revised. Further punctuation marks should be checking through the paper, especially after the equations and at the end of the statements.

* Remark, comments, and questions:

----- Title of paper is not clear. Try to clear meaning of the title.

------ The abstract is a little thin and does not quite convey the vibrancy of the findings and the depth of the main conclusions. The authors should please extend this somewhat for a better first impression.

------ The manuscript lacks motivation. Author needs to include the motivation of the study.

------Authors should write keywords in professional way.

----- There is already an abundance of modeling studies on COVID-19, vaccinations, and the months or years to come. However, apart from Ferguson's (now classic) work, Moore and Giordano, very little is said about similar modeling works. This is an issue for three reasons. First, the intended audience for such pieces is made of policy-makers and the general public: they are already facing an abundance of (occasionally conflicting) findings from models. If there is no attempt to contextualize the findings from this piece among others, then we're more likely to be adding noise to a crowded space, instead of providing valuable guidance. Second, several of the modeling assumptions made here may be in line with other pieces (which may provide some strength to the methods) or may be rather unique (which may need more discussion). Finally, as a piece of scientific literature, the contributions should be situated based on what already exists. In sum, the authors should explain how each of their assumptions and modeling choices compares to the literature; how their findings compare to the literature; and hence what is their specific contribution. Related models include, but are not limited to:

--https://doi.org/10.1007/s12190-021-01507-y

 ------Authors should insert all figures in appropriate places.

------Conclusion should be written in a more clear way. So try to short it and write in a professional way.

------Analysis is missing in paper so add it.

------Authors should improve the English of paper.

------Authors should correct grammatical error at few stage of paper.

-------Presentation of paper should be improved.

-------Try to reduce similarity of work.

-------References list are not appropriate.

Author Response

We appreciate the reviewer comments. As suggested by the reviewers, we have substantially modified our manuscript and provide below an answer to all comments.

Reviewer #2 (Comments for the Author):

The results listed in the paper in the form of formulas, figures, and analysis seems true and correct. The paper is well written and it is written in a truly sporty manner. English is generally good, I think it needs to be polished further and some typos need to be revised. Further punctuation marks should be checking through the paper, especially after the equations and at the end of the statements.

We appreciate the reviewer comments and thank the reviewer for his consideration.

A native English speaker has now edited the manuscript.

Remark, comments, and questions:

----- Title of paper is not clear. Try to clear meaning of the title.

We have proposed a new title: “Role and limits of COVID-19 vaccines in the delicate transition from pandemic mitigation to endemic control.”

------ The abstract is a little thin and does not quite convey the vibrancy of the findings and the depth of the main conclusions. The authors should please extend this somewhat for a better first impression.

Thanks for the comment. We have tried to somewhat expand the abstract in accordance with size constraint.

------ The manuscript lacks motivation. Author needs to include the motivation of the study.

We thank the reviewer for the question. We would like however to stress that our manuscript is not a study, it is a review with some aspect of personal views. The motivation is already described in the last section of the introduction. The public health choices and strategies put in place were mainly dictated by the solutions made available by the pharmaceutical industry. The scientific community was in some ways set aside. We think that the scientific community should be more involved to find solution to the current situation with the transition to an endemic circulation. We tried to make the wording clearer on our motivation.

------Authors should write keywords in professional way.

Although this comment is not clear to us, we have modified the keywords.

----- There is already an abundance of modeling studies onCOVID-19, vaccinations, and the months or years to come. However, apart from Ferguson's (now classic) work, Moore andGiordano, very little is said about similar modeling works. This is an issue for three reasons. First, the intended audience for such pieces is made of policy-makers and the general public: they are already facing an abundance of (occasionally conflicting) findings from models. If there is no attempt to contextualize the findings from this piece among others, then we're more likely to be adding noise to a crowded space, instead of providing valuable guidance. Second, several of the modeling assumptions made here may be in line with other pieces (which may provide some strength to the methods) or may be rather unique (which may need more discussion). Finally, as a piece of scientific literature, the contributions should be situated based on what already exists. In sum, the authors should explain how each of their assumptions and modeling choices compares to the literature; how their findings compare to the literature; and hence what is their specific contribution. Related models include,but are not limited to:

--https://doi.org/10.1007/s12190-021-01507-y

We thank the reviewer for this remark. We are not sure that this would be relevant to add more references on models, as our discussion is not based on models, but on observations of what occurred. A lot of models have been published, and a lot of conflicting data have been produced (as said by the reviewer) adding more confusion than insights. We made the choice of not basing our reflection on models. This is why, we did not add any reference on models.

Reviewer 3 Report

The work does not have a methodological part, which makes it difficult to fully evaluate it. It is not known what the purpose of the work is, what kind of work was done on obtaining the sources. Moreover, the topic suggests that the authors focus mainly on epidemiological aspects. If so, the topic is only slightly discussed, unless it is a prelude to considerations.

Despite wanting to evaluate the article, it is difficult for me to do so. An interesting and important topic, I do not want to attract the authors. But perhaps they would consider options for in-depth consideration, on a global scale, with specific research findings.

Author Response

We appreciate the reviewer comments. As suggested by the reviewers, we have substantially modified our manuscript and provide below an answer to all comments.

Reviewer #3 (Comments for the Author):

The work does not have a methodological part, which makes it difficult to fully evaluate it. It is not known what the purpose of the work is, what kind of work was done on obtaining the sources. Moreover, the topic suggests that the authors focus mainly on epidemiological aspects. If so, the topic is only slightly discussed, unless it is a prelude to considerations.

Despite wanting to evaluate the article, it is difficult for me to do so. An interesting and important topic, I do not want to attract the authors. But perhaps they would consider options for in-depth consideration, on a global scale, with specific research findings.

We appreciate the reviewer comments and thank the reviewer for his consideration.

We would like to stress that our manuscript is not a study, it is a review with some aspect of personal views. This is why our manuscript does not have any methodology section. We have modified the abstract to clarify this point.

Reviewer 4 Report

General comments

This is a timely review of COVID  vaccine strategy in the present phase of the pandemic. The authors discuss virological, epidemiological, and immunological aspects of the pandemic. They conclude that innovation is critical for control of the COVID-19 burden in the long term, by cooperation between universities, biotech and pharmaceutical companies.

The paper would benefit greatly from language editing by a native English speaker.

The authors use the term vaccine efficiency but I believe the authors should be using the term vaccine effectiveness.

https://www.gavi.org/vaccineswork/what-difference-between-efficacy-and-effectiveness?gclid=EAIaIQobChMI7v-ek-Tx-QIVsUeRBR31wAybEAAYASAAEgKS2vD_BwE

Omicron is spelled alternatively with capitalized and small o in different places. Please be consistent.

 Detailed comments:

Line 30-31. Is it correct to say that vaccination started in the early spring (in my terminology from March) or did it start in the winter (December- February)?

Line 31-32: the authors give the impression that the “eclipse” of COVID-19 in the summer of 2020 may have been caused by vaccination, perhaps unintentionally. Please rephrase to avoid misunderstanding as many other factors were at play, including non-pharmaceutical measures.

line 47: The authors write that the omicron variants BA 1-5  threaten the health care systems. While the number of patients admitted WITH COVID-19 is high, the virus causes less severe disease, and the authors’ statement should be a bit more nuanced. I do not believe the health care system in most countries are overburdened at this time, or perceive omicron as a great threat

Line 61: the authors only take into account future omicron variants, but I suggest that they also include new (non-omicrion) variants.

Line 65: “The emergence of a pandemic respiratory agent observed with SARS-CoV-2 is completely novel in the history of modern medicine.” I believe this is not quite true, We have had several influenza pandemics since 1957, the last in 2009, although I agree that all of them caused less severe disease than we have seen with COVID-19. Pleasy modify.

Line 93-94: “The most ambitious aim could at best be taking control of the virus circulation in some geographical regions, before controlling more globally its circulation”. How could this be done? How would you prevent the virus from spilling back from parts of the world where the virus continues to circulate in humans, and from animal reservoirs? Please modify or explain.

Line 117: in the fall of which year?

120-21: “Overall, the access of the vaccine should be globally improved as the hesitancy is growing [25].” This statement is unclear. How is vaccine access linked to vaccine hesitancy? I think the referenced article points to vaccine hesitancy as an additional challenge to lack of vaccine access. Please rephrase.

Line 143: the authors use the term air-borne disease. I believe a better term would be respiratory tract infections, which covers both infections transmitted by droplets and aerosol. Airborne is usually understood as transmitted by aerosol. (see also comment below)

Line 146-47: Is it true to say that SARS-CoV-2 has a life cycle limited to the lung? Please corroborate. What about other parts of the respiratory tract? And SARS-CoV-2 can be detected in blood and other organs. Is this not part of the life cycle?

Line: 152-154. What about adenovirus vector and other vaccine platforms that have been used? Perhaps earlier in the manuscript where mRNA vaccines are first mentioned. Please include at least short mention of other types of COVID-vaccines in use globally (although adenovirus vector mentioned in line 193).

Line 203: again the authors use the term “airborne”. I am unsure if the authors realize that this is still a contentious issue. Although it is widely acknowledged that aerosol transmission plays a role, many will argue that droplet transmission (in addition to contact) is the main mode of transmission. If the authors really mean to point out the importance of airborne (aerosol) transmission, I would suggest that they justify this by referring to studies or authoritative guidance that emphasizes this mode of transmission.

Author Response

We appreciate the reviewer comments. As suggested by the reviewers, we have substantially modified our manuscript and provide below an answer to all comments.

Reviewer #4 (Comments for the Author):

This is a timely review of COVID vaccine strategy in the present phase of the pandemic. The authors discuss virological, epidemiological, and immunological aspects of the pandemic. They conclude that innovation is critical for control of the COVID-19 burden in the long term, by cooperation between universities, biotech and pharmaceutical companies.

The paper would benefit greatly from language editing by a native English speaker.

The authors use the term vaccine efficiency but I believe the authors should be using the term vaccine effectiveness.

https://www.gavi.org/vaccineswork/what-difference-between-efficacy-and-effectiveness?

We appreciate the reviewer comments and thank the reviewer for his consideration.

We thank the reviewer for his suggestion, and the text has now been edited by a native English speaker.

We have now corrected the use of vaccine effectiveness rather than vaccine efficiency.

Omicron is spelled alternatively with capitalized and small o indifferent places. Please be consistent.

We have now corrected the spelling of Omicron and made it consistent throughout the text.

Detailed comments:

Line 30-31. Is it correct to say that vaccination started in the early spring (in my terminology from March) or did it start in the winter (December- February)?

Thanks for this remark. It started in the winter with very low supply of vaccines in most countries and the vaccine was available only for the most at risk populations. It, became massively available in the spring of 2021, with a chance to change the epidemiology of the outbreak. To be exact we corrected the text.

Line 31-32: the authors give the impression that the “eclipse” of COVID-19 in the summer of 2020 may have been caused by vaccination, perhaps unintentionally. Please rephrase to avoid misunderstanding as many other factors were at play, including non-pharmaceutical measures.

We agree and we have rephrased the sentence.

line 47: The authors write that the omicron variants BA 1-5 threaten the health care systems. While the number of patients admitted WITH COVID-19 is high, the virus causes less severe disease, and the authors’ statement should be a bit more nuanced. I do not believe the health care system in most countries are overburdened at this time, or perceive omicron as a great threat

We agree and we have rephrased the sentence.

Line 61: the authors only take into account future omicron variants, but I suggest that they also include new (non-omicron) variants.

We agree and we have rephrased the sentence.

Line 65: “The emergence of a pandemic respiratory agent observed with SARS-CoV-2 is completely novel in the history of modern medicine.” I believe this is not quite true. We have had several influenza pandemics since 1957, the last in 2009, although I agree that all of them caused less severe disease than we have seen with COVID-19. Please modify.

We agree and we have modified the text.

Line 93-94: “The most ambitious aim could at best be taking control of the virus circulation in some geographical regions, before controlling more globally its circulation”. How could this be done? How would you prevent the virus from spilling back from parts of the world where the virus continues to circulate in humans, and from animal reservoirs? Please modify or explain.

We agree and we have rephrased the sentence. This could be possible if we could get a vaccine blocking the virus transmission.

Line 117: in the fall of which year?

It was the fall of 2021. We have modified the sentence accordingly.

120-21: “Overall, the access of the vaccine should be globally improved as the hesitancy is growing [25].” This statement is unclear. How is vaccine access linked to vaccine hesitancy? I think the referenced article points to vaccine hesitancy as an additional challenge to lack of vaccine access. Please rephrase.

We agree and we have rephrased the sentence.

Line 143: the authors use the term air-borne disease. I believe a better term would be respiratory tract infections, which covers both infections transmitted by droplets and aerosol. Airborne is usually understood as transmitted by aerosol. (see also comment below)

We agree and we have rephrased the sentence.

Line 146-47: Is it true to say that SARS-CoV-2 has a life cycle limited to the lung? Please corroborate. What about other parts of the respiratory tract? And SARS-CoV-2 can be detected in blood and other organs. Is this not part of the life cycle?

We agree and we have rephrased the sentence.

Line: 152-154. What about adenovirus vector and other vaccine platforms that have been used? Perhaps earlier in the manuscript where mRNA vaccines are first mentioned. Please include at least short mention of other types of COVID-vaccines in use globally (although adenovirus vector mentioned in line193).

We agree and we have rephrased the sentence.

Line 203: again the authors use the term “airborne”. I am unsure if the authors realize that this is still a contentious issue. Although it is widely acknowledged that aerosol transmission plays a role, many will argue that droplet transmission (in addition to contact) is the main mode of transmission. If the authors really mean to point out the importance of airborne(aerosol) transmission, I would suggest that they justify this by referring to studies or authoritative guidance that emphasizes this mode of transmission.

We agree and we have rephrased the sentence.